# An unbiased template of the *Drosophila* brain and ventral nerve cord

**John A. Bogovic**[1]*, **Hideo Otsuna**[1], **Larissa Heinrich**[1], **Masayoshi Ito**[1], **Jennifer Jeter**[1], **Geoffrey Meissner**[1], **Aljoscha Nern**[1], **Jennifer Colonell**[1], **Oz Malkesman**[1], **Kei Ito**[2], **Stephan Saalfeld**[1]*

**1** Janelia Research Campus, Howard Hughes Medical Institute, Ashburn, Virginia, United States of America, **2** Institute of Zoology, University of Cologne, Germany

* bogovicj@janelia.hhmi.org (JB); saalfelds@janelia.hhmi.org (SS)

**Data Availability Statement:** Data are available through figshare. Links to each dataset uploaded to figshare are found here: https://www.janelia.org/open-science/jrc-2018-brain-templates and in the manuscript.

## Abstract

The fruit fly *Drosophila melanogaster* is an important model organism for neuroscience with a wide array of genetic tools that enable the mapping of individual neurons and neural subtypes. Brain templates are essential for comparative biological studies because they enable analyzing many individuals in a common reference space. Several central brain templates exist for *Drosophila*, but every one is either biased, uses sub-optimal tissue preparation, is imaged at low resolution, or does not account for artifacts. No publicly available *Drosophila* ventral nerve cord template currently exists. In this work, we created high-resolution templates of the *Drosophila* brain and ventral nerve cord using the best-available technologies for imaging, artifact correction, stitching, and template construction using groupwise registration. We evaluated our central brain template against the four most competitive, publicly available brain templates and demonstrate that ours enables more accurate registration with fewer local deformations in shorter time.

## 1 Introduction and related work

Canonical templates (or "atlases") of stereotypical anatomy are vital in inter-subject biological studies. In neuroscience, atlases of the central nervous system have become an important tool for cumulative and comparative studies. Neuroanatomical templates for humans [1, 2], mouse [3, 4], C. *elegans* [5], and *Drosophila* [6–8] have been valuable and influential for studies of anatomy, function, and behavior [7, 9–11].

For a template space to be useful, it must be possible to reliably find spatial transformations between individual subjects and that template. The transformation must be capable of expressing the biological variability so that stereotypical anatomical features of many transformed subjects are well aligned in template space. This serves to normalize for "irrelevant" sources of variability while comparing across a population. Early templates used simple affine spatial transformations [1], for which identifying a small number of stereotypical landmarks was sufficient. More recently, image registration has enabled the computation of more flexible (elastic, diffeomorphic, etc.) [12] transformations to be found between pairs of images. As a result, modern templates consist of a representative digital image of the anatomy and imaging

**Funding:** This work was funded by the Howard Hughes Medical Institute.

**Competing interests:** The authors have declared that no competing interests exist.

modality of interest, which is used as the target (or "fixed" image) for image registration, which generates the spatial transformation [13].

Transforming individual subjects' anatomy to a template space enables the comparison and analysis of a cohort of individuals in a common reference space. Typical registration approaches are either feature/landmark-based, as in Peng et al. [14], or pixel based, using generic image registration software libraries, such as ANTs [15], CMTK [16], or elastix [17]. If the template itself has anatomical labels superimposed, then it also enables an individual's anatomy to be labeled via the spatial transformation to the template [18].

The fruit fly *Drosophila melanogaster* is an important model organism for neuroscience. The availability of powerful genetic tools that enable precise imaging and manipulation of specific neuronal populations [19–21] have made possible many important biological experiments. For example, the Fly-circuit database by Chiang et al. [22] reconstructed approximately 16,000 neurons in *Drosophila* from light microscopy. Jenett et al. [6] produced and imaged 6,650 GAL4 lines that have enabled the cataloging of a wide array of neurons in the *Drosophila* central nervous system. Among other advances, these resources enabled the creation of a spatial map of projection neurons for the lateral horn and mushroom body of *Drosophila* [23]. Panser et al. [10] generated a spatial clustering of the *Drosophila* brain into functional units on the basis of enhancer expression. Using over 2,000 GAL4 lines, Robie et al. [11] created a whole brain map linking behavior to brain regions. Yu et al. [24] explored *Drosophila* development, analyzing the connectivity of neuronal lineages by neuropil compartment.

Canonical templates and spatial alignment are an important aspect of the above studies that leverage genetic tools and imagery in *Drosophila*. Any given line labels a different sub-population of neurons and so it is necessary to perform comparisons in a canonical space. Many brain templates exist for *Drosophila*, a summary of which we give below.

Rein et al. [25] generated a template starting with 28 individual brains, stained with nc82 [26] and imaged at $0.6 \times 0.6 \times 1.1$ μm resolution. The template brain was chosen as the individual with "the average volume for each substructure." Registration was done by first estimating a global rigid, then estimating a per-structure rigid or similarity transformation which was interpolated over space.

Jenett et al. [6] selected a representative confocal image of a female fly brain, imaged at $0.62 \times 0.62 \times 1.0$ μm/px as a standard brain for their GAL4 driver line resource, and was later resampled in *z* to an isotropic resolution of 0.62 μm/px. We will refer to this template as JFRC 2010.

Aso et al. [7] selected another single female brain for their work, called the JFRC 2013 template. It comprises five stitched tiles, imaged at $0.19 \times 0.19 \times 0.38$ μm/px, then downsampled to an isotropic resolution of $0.38 \times 0.38 \times 0.38$ μm/px. Since the JFRC 2010 and JFRC 2013 templates consist of a brain image of a single individual fly, we will call these "individual templates."

Ito et al. [27] introduced the "Ito half-brain" as their reference standard. It includes a rich set of compartment labels for neuropil boundaries and fiber bundles but automatic registration of whole brains to this standard is not straightforward.

The female, male, and unisex FCWB templates by Ostrovsky et al. [28] were generated from images manually selected from the Fly-Circuit database [22], imaged at $0.32 \times 0.32 \times 1.0$ μm/px, using groupwise registration [29] with the CMTK registration software. Groupwise registration is the process of co-registering a set of images without specifying one particular image as the registration target and thereby avoids bias (see Section 2.4 for a more detailed description). Seventeen brain samples were used for the female template and nine for the male template. The two gendered average brain templates were themselves registered and averaged with equal weight to create a unisex template.

Arganda-Carreras et al [8] recently used groupwise registration with the ANTs registration software to generate an improved unbiased *Drosophila* brain template from ten individual fly brains, imaged at $0.6 \times 0.6 \times 0.98$ μm/px and labeled with the nc82 antibody. We call this the "Tefor" template. To measure registration performance, the authors compare overlap of anatomical labels of individuals after registration, and conclude that templates generated by groupwise registration outperform templates consisting of an individual brain image.

Given the abundance of brain templates for *Drosophila*, Manton et al. [30] created "bridging transformations" that align many of these templates, including JFRC 2010, JFRC 2013, FCWB, and the Ito half-brain. Bridging transformations link previously disparate data-sets and thereby enable comparisons across all datasets in the space of any of these templates. The combination of neuronal database, image registration, and brain template has made neuron matching with NBLAST [31] an important technology. NBLAST has also recently found success matching neurons obtained from different, complementary data-sets [32], including the first complete electron microscopy (EM) volume of the female adult fly brain (FAFB) generated by Zheng et al. [33]. This image volume enables the complete tracing of every neuron and identification of synapses spanning the central brain in a single individual, and is potentially a very useful reference standard brain.

Meinertzhagen et al. [34] describes how neurons identified from LM will offer an important form of validation for neurons traced in EM, either manually or automatically. This capability depends on spatial alignment between the two modalities. A plugin for Fiji [35] called ELM (github.com/saalfeldlab/elm) is a custom wrapper of the BigWarp plugin [36], and was used to manually place landmark point correspondences between the LM template and generate a spatial transformation from the EM image space to the JFRC 2013 template space.

Using this registration, Zheng et al. [33] showed that neurons traced in FAFB can be matched with neurons cataloged from light microscopy (LM), a capability that will enable researchers to simultaneously leverage the advantages of each modality: dense connectivity from EM, and cell type, neurotransmitter, gene expression, etc. information from LM [32].

While studies of the *Drosophila* ventral nerve cord (VNC) are numerous and ongoing [37–40], efforts in generating a standard anatomical coordinate system are somewhat lacking. Borner et al. [41] created a standard *Drosophila* VNC, but the data are not available in a public respository, thus its impact is limited. Recently, Court et al. [42] developed a standard nomenclature for the ventral nervous system.

In summary, all existing *Drosophila* brain templates lack certain desirable characteristics. Some consist of a individual samples and are therefore biased. Those that use groupwise registration to avoid bias use a small number of subjects, imaged at relatively low resolution and do not leverage new advances in tissue preparation, imaging, and artifact correction. No groupwise averaged ventral nerve cord template exists in a public repository.

## 1.1 Contributions

We generated unbiased, symmetric, high-resolution, male, female, and unisex templates for the *Drosophila* brain and ventral nerve cord using the newest advances in tissue preparation, imaging, artifact correction, and image stitching. The images of individual samples that comprised the template were acquired at high resolution of $0.19 \times 0.19 \times 0.38$ μm/px. We used groupwise image registration to ensure that the shape of the resulting template is not biased toward our choice of subject. Fig 1 shows the female, unisex, and male templates for the *Drosophila* central brain and ventral nerve cord. We call this set of templates "JRC 2018."

We performed a thorough comparison using the four most competitive publicly available *Drosophila* brain templates, and three leading image registration software libraries, measuring

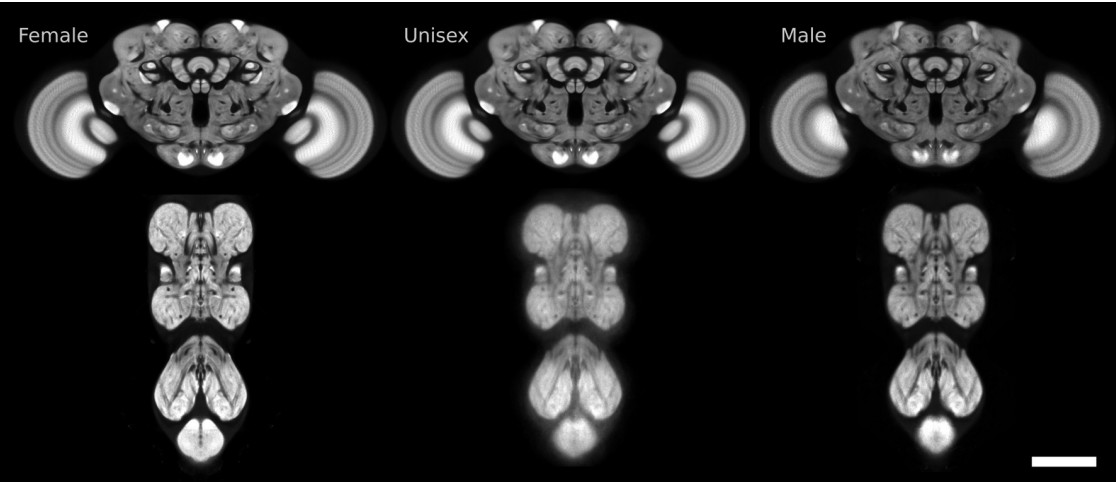

**Fig 1. Slices of the six templates we created for female, unisex, and male *Drosophila* central brains and ventral nerve cords.**
Scale bar 100 μm.

registration quality, amount of deformation, and computational cost. Each template was evaluated using eight different choices of registration algorithms/parameter settings. We show that *Drosophila* brain samples register significantly *better*, *faster*, and with *less local deformation* to our JRC 2018 template than to any prior template, enabling more accurate comparison studies than were previously possible.

We also generated a new, automatic registration between the female full adult fly brain (FAFB) electron microscopy data-set of Zheng et al. [33] using automated synapse predictions [43]. This registration is more data-driven, less likely to suffer from variable error, and potentially better regularized than the existing manual registration generated with ELM [33, 36].

We developed software to create, apply, convert, and compare templates and transformations. Our software depends on the publicly available registration packages elastix, CMTK, and ANTs. Our specific contributions are:

1. Scripts and parameters for template construction (these are customized versions of scripts from ANTs).

2. Scripts and parameters for registration used in evaluation and analysis, including registration quality estimation (see Section 3.2.1).

3. Software for applying transformations to images, skeletons, and sets of point coordinates. Both command line utilities and Fiji [35] plugins are available.

4. A new compressed HDF5 based format (see Supplementary Notes A.8 in S2 File) for multi-scale transformations and conversion tools between this new format (including quantization and downsampling) and transformations generated by the registration packages elastix, CMTK, and ANTs.

The templates and transformations can be found on-line at https://www.janelia.org/open-science/jrc-2018-brain-templates. Software and code supporting these resources are available at https://github.com/saalfeldlab/template-building.

**1.1.1 Usage.** This section outlines common use cases with pointers to data, transformations, code, and demonstrations. The most common use case will be for *Drosophila* neuroscience researchers seeking to combine or compare neuronal reconstructions or spatial data

across datasets. This includes transforming neuropil labels across datasets or from a template to an individual brain. It also enables the matching of neurons reconstructed from electron microscopy to those in a light microscopy dataset, or vice versa, with NeuronBridge [44, 45]. Our supporting code can be used to apply transformations between templates to raw points or neurons stored as swc files. The NeuroAnatomy Toolbox [46] also integrates this template and the associated bridging transformations.

Researchers collecting imaging data can register their data to this template with the goals of spatially aligning image data to each other and to other templates or datasets. In this case, uses should obtain the template image and registration scripts and parameters. Registration algorithm parameters may need adjusting depending on how similar the collected image data are to those tested in this work.

## 2 Materials and methods

### 2.1 Sample preparation and image acquisition

**2.1.1 Template data.** All samples were based on the transgene brp-SNAP [47] and labeled with 2 μM Cy2 SNAP-tag ligand [48]. Samples were fixed for 55 minutes in 2 temperature, then fixed for 4 hours in 4 Samples were dehydrated in ethanol, cleared in xylene, and mounted in DPX, as described at https://www.janelia.org/project-team/flylight/protocols [7, 49].

Samples were imaged unidirectionally on six Zeiss 710 LSM confocal microscopes with Plan Apo 63 × /1.4 Oil DIC 420780/2-9900 lenses, at a resolution of 0.19 μm/px, in tiles of 1024 × 1024 px (192.6 × 192.6 μm), a pixel dwell time of 1.27 μm, a *z*-interval of 0.38 μm, and a pinhole of 1 AU to 488 nm. Brains were imaged in five overlapping tiles, VNCs in three. Scanning was controlled by Zeiss ZEN 2010 software and a custom MultiTime macro, as described by Jenett et al. [6].

Tiles were corrected for lens-distortion and chromatic aberration (see Section 2.2) and stitched with Fiji's [35] stitching plugin [50]. Custom scripts were used to parallelize processing on the Janelia CPU cluster. For template construction, 62 central brains (36 female) and 75 ventral nerve cords (36 female) were acquired.

**2.1.2 Evaluation data.** For evaluation, we chose 20 female flies imaged with both an nc82 channel and a channel in which neuronal membrane of a split GAL4 driver line were labeled with a myristoylated FLAG reporter, as described by Aso et al. [7]. We selected four split GAL4 lines that label neurons with broad arborization, that together, cover nearly the whole brain. Maximum intensity projections of the neuronal membrane channel for these lines are shown in Fig 2. Our testing cohort consisted of 20 individuals in total, summarized in Table 1. As a result, our evaluation includes measurements across the whole brain and does not focus on a particular subset of the anatomy.

### 2.2 Lens distortion and chromatic aberration correction

**2.2.1 Re-usable calibration slides.** Measurements to establish and update the calibration models were made by imaging multicolor beads mounted, like the tissue samples, on #1.5H cover slips. The beads are 1 μm silica (Polysciences 24326-15), functionalized through reaction with tri-ethoxy-sliane-PEG biotin (Nanocs #PEG6-0023). The biotinylated surface was labeled with a mixture of dye-labeled streptavidin. The dyes were Alexa 488 (ThermoFisher S32354), Cy3 (Jackson Immunoresearch 016-160-084), Alexa 594 (ThermoFisher S32356), and Atto647N (Sigma-Aldrich 94149-1MG). The beads were deposited out of tris-buffered saline onto plasma cleaned coverslips, dried, and mounted in DPX according to the same tissue mounting protocol described above.

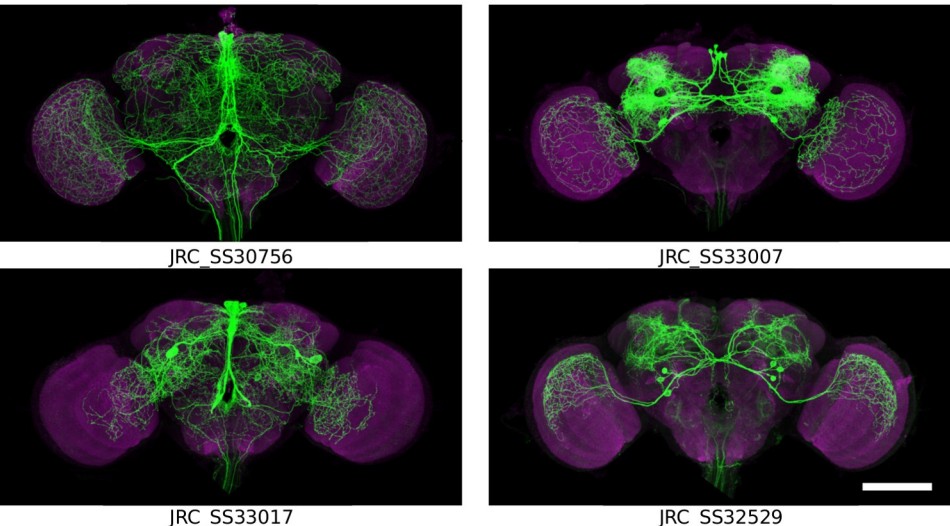

**Fig 2. Maximum intensity projections of individuals from four of the GAL4 driver lines used to evaluate registration accuracy (magenta nc82, green GAL4).** Notice the broad arborization spanning most of the central brain and optic lobes. Scale bar 100 μm.

**2.2.2 Distortion correction.** Image stack mosaics of 4 × 4 tiles with 50–60 taken using the same settings as described above. Image stacks were acquired in two passes, first at 488 nm and 594 nm, then at 488 nm, 561 nm, and 647 nm.

Channels were separated and max-intensity projections for each single channel tile were created with a custom Fiji script. Single channel 4 × 4 mosaics were imported as individual layers in TrakEM2 [51]. Channel mosaics were pre-stitched to account for stage shift. A non-linear lens-distortion correction model for each channel was estimated with a new extended version of the method by [52] that we made available in TrakEM2. Lens-corrected channel mosaics were stitched and then all channels were globally aligned with an affine transformation model. The composition of the non-linear lens-correction model and the affine alignment transformation correct for both lens-deformation and chromatic aberration. These correction models were generated and exported, for all confocal microscopes (five Zeiss LSM 710's and one Zeiss LSM 780). Finally, they were applied to individual 3D image stacks of *Drosophila* brain samples prior to stitching.

We recorded detailed video instructions to reproduce the calibration protocol and made them available on Youtube: https://www.youtube.com/watch?v=lPt-WQuniUs. Code can be found on-line at https://github.com/saalfeldlab/confocal-lens.

**Table 1. Split GAL4 driver lines chosen for evaluation with the count of the number of subjects per line (#).**

| Line | | # |
|---|---|---|
| JRC_SS23204 | GMR_13F04-p65ADZp in attP40 and GMR_13B05-ZpGDBD in attP2 | 2 |
| JRC_SS23206 | GMR_28F06-p65ADZp in attP40 and GMR_13B05-ZpGDBD in attP2 | 2 |
| JRC_SS23208 | GMR_40F04-p65ADZp in attP40 and GMR_13B05-ZpGDBD in attP2 | 2 |
| JRC_SS30756 | GMR_15C12-p65ADZp in attP40 and GMR_64B11-ZpGDBD in attP2 | 3 |
| JRC_SS30777 | GMR_70D06-p65ADZp in attP40 and GMR_64B11-ZpGDBD in attP2 | 2 |
| JRC_SS33007 | GMR_54H04-p65ADZp in attP40 and GMR_10D10-ZpGDBD in attP2 | 3 |
| JRC_SS33017 | GMR_67A07-p65ADZp in attP40 and GMR_35G08-ZpGDBD in attP2 | 3 |
| JRC_SS32529 | GMR_21G11-p65ADZp in attP40 and GMR_78H08-ZpGDBD in attP2 | 3 |

## 2.3 Neuron skeletonization

We computed skeletons from the neuron channel by first applying 3D direction-selective local-thresholding (DSLT) [53] to each raw image tile. DSLT convolves the image with multiple scaled and rotated cylindrical kernels. We used kernel radii of 2, 6, and 10 voxels. The maximum response was thresholded to give a neuron mask. This mask was transformed along with the tile during image stitching, and then skeletonized using the 3D Skeletonization plugin in Fiji [35].

## 2.4 Template construction

We constructed our template using groupwise registration, which seeks to find both an average shape and average intensity across all individuals in a cohort and is described in Avants et al. [29, 54] The script `buildtemplateparallel.sh` that is part of the ANTs library, implements this. Our modifications, described below, can be found on-line at https://github.com/saalfeldlab/template-building.

Groupwise registration begins with an initial template—often a single individual or the average of the unregistered cohort of images is selected (we chose the latter). Next, every individual image is registered to that initial template using a particular registration algorithm (transformation model, similarity measure, optimization scheme) and then transformed to the template space. Next, the set of transformed individual images are averaged and the transformations are averaged. This yields new mean intensities as well as a mean transformation from each subject to the current template estimate. Finally, the new mean intensity image is transformed through the inverse of the mean transformation to obtain a new template. This procedure of registration-averaging-transformation is iterated to transport the initial template toward the mean intensity and shape.

We made several changes seeking to reduce the amount of deformation possible during registration. First, we used the elastic transformation model rather than SyN diffeomorphic model [15]. Second, we regularized the transform more strongly, the details of which can be found in our open source repository. We sought to generate a template with left-right symmetry. Similar to the approach used in the in Allen common coordinate framework [3], we left-right flipped every individual brain and VNC in our cohort and included both the original and flipped images in the groupwise registration procedure. This doubled both our effective image count and computational expense during groupwise registration. We discuss this choice further in Section 4.5.

We downsampled the raw and left-right flipped images to 0.76 μm/px isotropic resolution prior to running groupwise registration in order to reduce computational cost/runtime. We believe that this downsampling did not have an appreciable effect on the template construction. This is supported by experiments we performed showing that lower resolution templates can have similarly good performance as high resolution templates (see Supplementary Notes A.4 in S2 File). The final template was obtained by applying the transformation computed at low resolution to the original, high resolution images, resulting in a high resolution template. All templates were rendered at 0.19 μm/px isotropic resolution. This equals the *xy*-resolution of the original images, but has a higher *z*-resolution (by about a factor of 2). We chose this primarily for convenience, as the only downside is the additional storage space.

We applied this procedure to different subsets of our image data to produce female, male and unisex templates. For the central brain, we used 36 female individuals (72 images including left-right flips) for the female template, 26 male individuals (52 image with left-right flips) for the male template, and the union of both for the unisex brain template: 62 individuals (124 images with left-right flips). For the ventral nerve cord we used 36 female individuals (72

images including left-right flips) for the female template, 39 male individuals (78 image with left-right flips) for the male template, and the union of both for the unisex VNC template: 75 individuals (150 images with left-right flips).

## 2.5 Bridging registrations

Transformations between different template spaces have proven useful in neuroscience because they unify disparate sets of data that are each aligned to different templates [30]. We computed forward and inverse bridging registrations between our template and the other templates we evaluated in this work: JFRC 2010, JFRC 2013, FCWB, and Tefor. We used the algorithm ("ANTs A") that we found performs best overall (see Supplementary Notes A.2 in S2 File). In Fig 3, we show these templates in the space of JRC 2018 and vice versa. Qualitatively, these registrations appear to be about as accurate as those from individuals to templates. We used these transformations to normalize distances measured across templates (see Section 3.2).

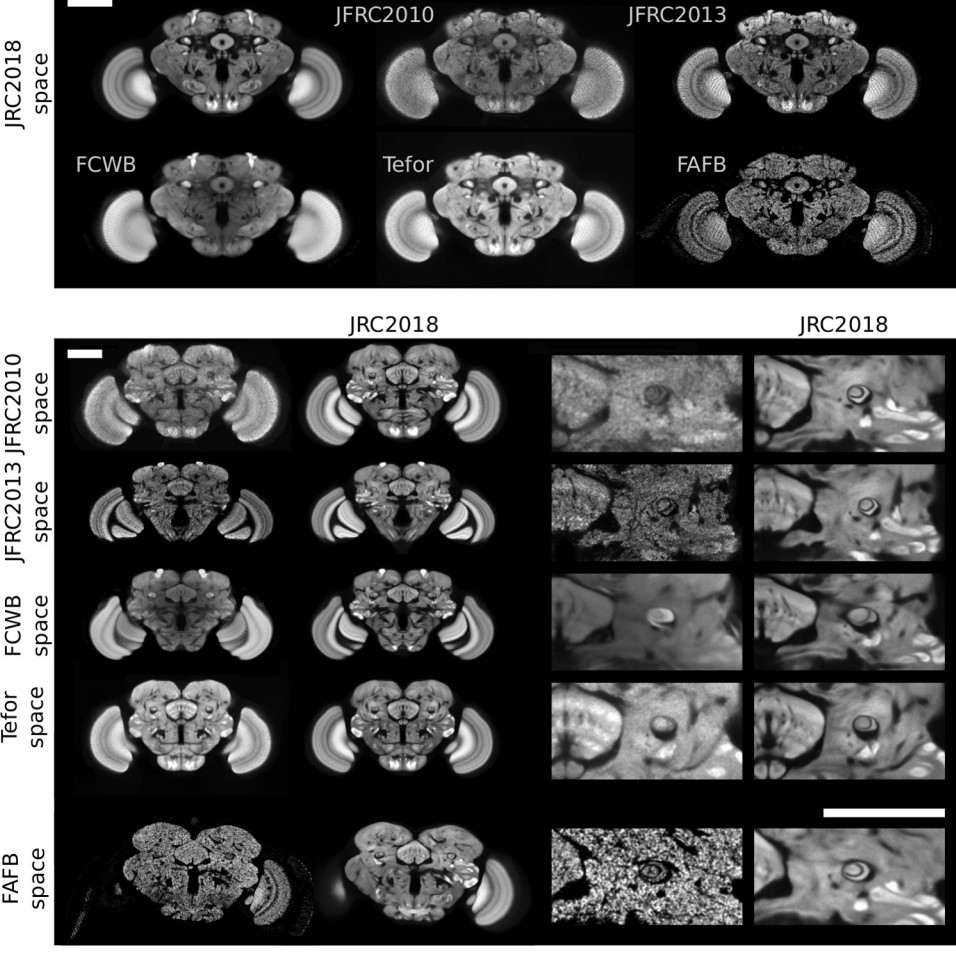

**Fig 3. Visual comparison of *Drosophila* brain templates and bridging transformations.** The top two rows show four existing templates registered to our JRC 2018 female template, as well as synaptic cleft predictions derived from the FAFB EM volume, transformed into the space of JRC 2018F. The middle four rows show JRC 2018F (second and fourth columns) registered to each of the three templates, along with a close-up around the fan-shaped body and the pedunculus of the mushroom body. The bottom row shows JRC 2018F transformed into the space of FAFB. Scale bars 100 μm.

We also provide transformations between the unisex template and the male and female templates to facilitate inter-sex comparisons.

## 2.6 Registration with electron microscopy

We automatically aligned the FAFB EM volume and the JRC 2018 female template (see Fig 3). We rendered and blurred (with a 1.0 µm Gaussian kernel) the synapse cleft distance predictions generated by Heinrich et al. [43] at low resolution (1.02 × 1.02 × 1.04 µm) so that the resulting image has an appearance similar to an nc82 or brp-SNAP labeled confocal image. We used ANTs A to find a transformation between the two images, the result of which produces a qualitatively accurate alignment. In Supplementary Notes A.7 in S2 File, we discuss additional motivation and tradeoffs, including additional visualizations and a quantitative evaluation. In particular, we show that this transformation reproduces scientific results by Zheng et al. [33]. The accuracy of the manual registration between FAFB and JFRC 2013 by Zheng et al. [33] suffers from variable error caused by the preference of the human annotator for specific brain regions and spurious details. Our automatic registration does not have this preference and is potentially better regularized.

More recently, Xu et al. [55] released the *Drosophila* "hemibrain" dataset, a focused ion bean scanning electron microscopy (FIBSEM) image volume, along with neuronal reconstructions. We have also created a transformation from our template to the hemibrain dataset and made it publicly available. The process we used to create the transform for the hemibrain is similar to that for FAFB. We processed and downsampled a synapse prediction [55] to create a synthetic synapse density image, which was registered to the template as above. Registration was somewhat more challenging for FAFB, given that the the hemibrain image does not cover the entire brain. For that reason, automatic registration was followed by a manual correction of the transformation using Bigwarp [36]. This bridging transformation enabled the matching of segmented neurons from the hemibrain dataset to neurons imaged with light microscopy using NeuronBridge [44, 45] or the natverse [46].

## 3 Results

We compared the registration tools ANTs [15], CMTK [16], and elastix [17] using three sets of parameters for ANTs and CMTK and two sets of parameters for elastix, giving a total of eight different registration algorithms. We used images from 20 female flies for evaluation, with details described in Section 2.1.2. Since all testing subjects were female, we used the female JRC 2018 template for the experiments below, but will refer to it as simply "JRC2018F."

## 3.1 Qualitative comparison

In Fig 3, we visually compare slices through the JFRC 2010 [6], JFRC 2013 [7], FCWB [28], Tefor [8], and the JRC 2018F brain templates. Note the improved contrast and sharpness of anatomical structures in our JRC 2018F template relative to the others. We describe possible reasons for this in the Section 4.

Fig 4 shows *xz*-slices through the five templates and FAFB, where the *x*-axis is medial-lateral and the *z*-axis is anterior-posterior. Confocal imaging results in poorer *z*- than *xy*-resolution: 0.38 µm/px vs. 0.19 µm/px for our images. The brain templates vary notably in physical size and resolution (see Table 2). Specifically, the FCWB and JFRC 2010 templates are physically much smaller in the anterior-posterior direction. The affine part of the bridging transformations computed for Fig 3 indicate that the FCWB template is about 40.

We make additional qualitative comparisons available in the S1 File. S1 Fig in Supplementary Notes A.1 in S2 File shows registered images overlaid on each template for one choice of

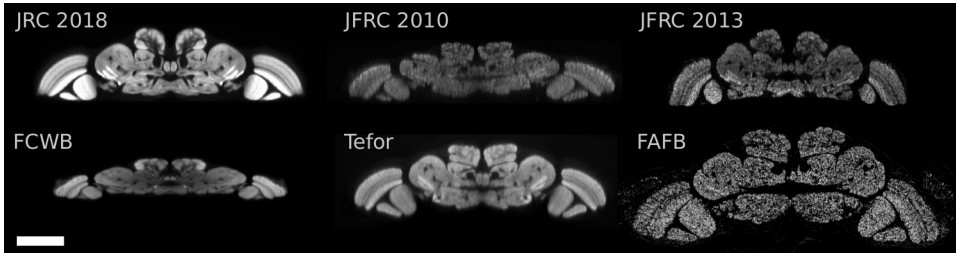

**Fig 4. Horizontal (*xz*-slice) visualization of five brain templates and FAFB in physical coordinates.** Note that the lower *z*-resolution is appreciable for individual templates (JFRC 2010 and JFRC 2013). Furthermore, observe the significant differences in physical sizes across these brain templates. Scale bar 100 μm.

registration algorithm. Supplementary Notes C in S2 File contains much more detail, showing registration results for every template and algorithm.

## 3.2 Quantitative comparison

Measuring and evaluating the accuracy of image registration is notoriously challenging [56]. Common measures include image similarity, distance between landmark points, or overlap of semantic (anatomical) regions, each of which come with their own disadvantages. Image similarity can suffer from bias because the measurement may be the same quantity the registration algorithm is optimizing for (or a correlated quantity) and does not account for implausible deformation. Landmark points measure accuracy sparsely and are costly to generate (since they are usually manually placed). Region overlap is even more costly and only sensitive to differences along or near boundaries of those regions. We discuss this in more detail in Section 4.1.

**3.2.1 Registration accuracy measure.** We measure registration accuracy with a naïve per-node skeleton distance between the same neuronal arbor for different individual flies after registration to the template and normalizing for template size. Normalizing for size and shape is necessary given the observations from Fig 4 that some templates are physically smaller than others. Transforming skeletons to a physically smaller space would artificially decrease the distance measure. Similarly, if shape differs, regions with locally dense distributions of arbors could be compressed, leading to smaller distances for many point samples.

We compute this distance using the following procedure. Neurons are skeletonized using the methodology described in Section 2.3. Nc82 channels for two *Drosophila* individuals are independently registered to a template. The transformation found using the nc82 channel is then applied to the neuronal skeleton to bring them into template space. This is followed by an additional normalizing transformation that ensures all templates are at an equivalent scale and shape, that of JRC 2018F (see Section 2.5). The distance transform of both skeletons is computed and rendered at an isotropic resolution of 0.5 μm/px. For a given point (pixel) on one skeleton, the value of the distance transform of the other skeleton gives the naïve orthogonal

**Table 2. The resolutions of all templates in μm/pixel.**

|     | JRC 2018F | JFRC 2010 | JFRC 2013 | FCWB | Tefor | FAFB |
|-----|-----------|-----------|-----------|------|-------|------|
| *xy* | 0.19 | 0.62 | 0.62 | 0.32 | 0.61 | 0.004 |
| *z* | 0.19 | 0.62 | 0.62 | 1.0 | 0.99 | 0.04 |

Evaluation against our JRC 2018F template was performed at a resolution of 0.62 μm/px or less. We added the resolution of the FAFB synapse cloud for comparison.

**Table 3. Mean (Standard deviation) of skeleton distance by template, in μm.**

| Template | Best per template | | Fixed algorithm | | |
| --- | --- | --- | --- | --- | --- |
| | | Algorithm | ANTs A | CMTK A | Elastix A |
| JRC2018F | 3.97 (3.65) | ANTs A | 3.97 (3.65) | 4.03 (3.70) | 4.05 (3.73) |
| Tefor | 4.00 (3.68) | ANTs A | 4.00 (3.68) | 5.44 (5.75) | 4.12 (3.69) |
| JFRC2013 | 4.06 (3.68) | Elastix A | 4.15 (3.84) | 4.73 (4.71) | 4.06 (3.68) |
| JFRC2010 | 4.11 (3.68) | Elastix A | 6.31 (6.20) | 5.69 (5.50) | 4.11 (3.68) |
| FCWB | 4.35 (3.97) | ANTs A | 4.35 (3.97) | 5.52 (4.97) | 6.18 (5.93) |

The header spans "Mean (standard deviation) skeleton distance (μm)".

Templates are ordered by decreasing performance for each template's best algorithm (given in the third column). Lower distances indicate better registration performance. The three rightmost columns show statistics of skeleton distance for all templates when fixing the algorithm, where we choose one set of parameters for each registration library.

distance between the skeletons. We report statistics of this skeleton-distance (in μm) both over the entire brain and split by compartment labels defined by JFRC 2010. Image processing was done using custom code based on ImgLib2 [57]. Statistics, analysis and visualization used pandas [58] and matplotlib [59]. All evaluation code can be found on-line at https://github.com/saalfeldlab/template-building.

The skeleton distance is similar to landmark distance, but involves no manual human decision making or interaction since the structures of interest are directly specified by the anatomy. Our approach has the advantage of being fast and simple to compute as well as providing a distance for every point on the skeleton. It is limited in that it only measures distances perpendicular to the skeletons and assumes strict correspondence between the nearest points on two skeletons. It will therefore tend to underestimate errors. See Section 4.1 in Section 4 for more details on the benefits and limitations of this performance measure.

We measured the skeleton distance between all pairs of individuals belonging to the same line, though we grouped the first five lines in Table 1 (JRC_SS23204, JRC_SS23206, JRC_SS23208, JRC_SS30756, JRC_SS30777) since they have very similar expression patterns. There exist 110 permutations of the 11 flies in the first line-group and 6 permutations each of the other three lines, making 128 pairwise comparisons in total. This distance is computed at every pixel for a given skeleton pair. When rendered at 0.5 μm resolution, each skeleton comprises about 250,000 pixels on average, yielding about 32 million points for which distance is estimated across all 128 skeleton pairs. This is repeated for each template and algorithm pair we evaluated.

We then compute several statistics of the distance across pairs of skeletons with a fixed registration algorithm and template. Table 3 dist template shows the mean and standard deviation of the distance distribution for different templates and algorithms. JRC 2018F has the lowest mean values of distance (indicating best performance) both when averaging across algorithms, and when choosing the best algorithm for a given template. Figs 5 and 6 plot skeleton distance against a deformation measure (see Section 3.2.2) and CPU time, respectively. We omit p-values here because their magnitude is largely a reflection of our large sample size (all pairwise hypothesis tests are significant). Table 3 dist template shows the mean and standard deviation of skeleton distances averaged across 128 pairs of flies (four driver lines). The second column gives the mean distance averaged also across eight choices of registration algorithm/parameters, while the third and fourth columns give average distances for the "best" two algorithms (see S1 Table in S2 File). The JRC 2018F template performs best according to this measure, independent of the registration algorithm used.

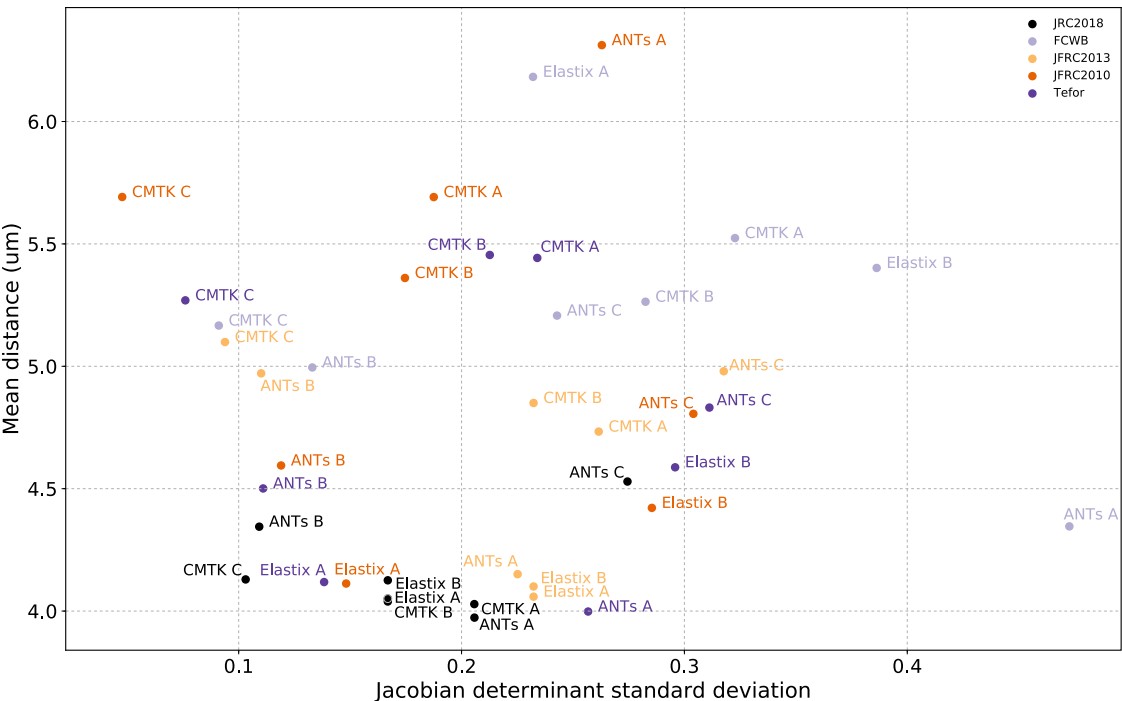

**Fig 5. Scatterplot showing mean skeleton distance and standard deviation of the Jacobian determinant for all template-algorithm pairs.**

The statistics we report in this section are computed over the whole brain. Supplementary Notes B in S2 File contains tables with additional statistics computed over subsets of the brain defined by 76 compartment labels.

**3.2.2 Deformation measure.** In addition to measuring accuracy, we also report the standard deviation of the Jacobian determinant (JSD), a scalar measure that describes the deformation or distortion of a brain's shape after undergoing a transformation. This is important if downstream analyses in template space rely on anatomical morphology, see Section 4.3 for a discussion. The Jacobian determinant of a transformation at a particular location describes the amount of local shrinking/stretching, where a value of 1.0 indicates that volume is locally preserved, less that 1.0 indicates volume decrease, and greater than 1.0 indicates volume increase. The standard deviation of the distribution of the Jacobian determinant map reflects deformation because the spread of this distribution (not the mean) captures the extent to which a transformation is *simultaneously* stretching and shrinking space. A similarity transformation will have a fixed value of the Jacobian determinant equal to its scale parameter at all points. As a result, its mean Jacobian determinant will equal that value. Therefore, the mean value of the Jacobian determinant is not indicative of a deformation, but rather shows average scaling. In the supplement, S4 Table in S2 File shows that the *mean* value of of the Jacobian determinant is very nearly 1.0 for most templates, as we would expect. We computed the JSD from displacement fields and do not include the affine component to avoid confusing global scale changes with deformation. We consider an alternative measure of deformation in Section 3.2.3.

In Fig 5, we plot the JSD against the mean skeleton distance described above. We observed that algorithms with stronger regularization (CMTK C, ANTs B) have a lower JSD, as we would expect. The mean distance between skeletons is lowest for the JRC 2018F template. Using the best overall algorithm (ANTs A), the deformation when using JRC 2018F is lower

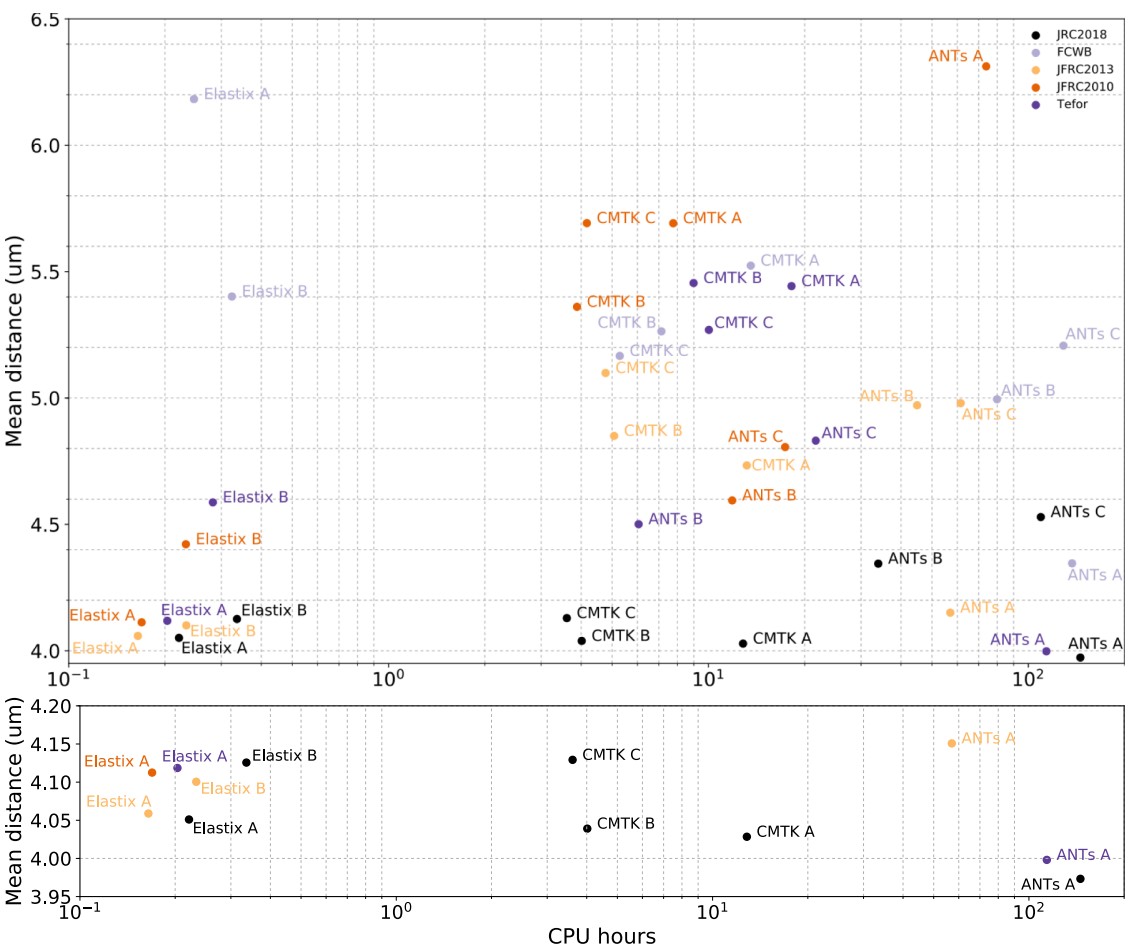

**Fig 6. Scatterplot showing mean skeleton distance and the mean computation time in CPU-hours for all template-algorithm pairs (above), and the best performing pairs (below).**

than when using Tefor, the next best template. Among the fastest algorithms (Elastix A and Elastix B), the JSD for JRC 2018F is lower than that for JFRC 2013, but higher than Tefor and JFRC 2010.

**3.2.3 Another deformation measure.** In Section 3.2.2, we explain that the standard deviation of the Jacobian determinant over space measures deformation. One reason to use that measure is that the registration libraries we tested here (ANTs, CMTK, elastix) provide functions to compute the Jacobian determinant. Despite its prevalent use, it may not always measure the kind of deformation researchers care to avoid.

For example, a transformation with regions of volume decrease and regions of volume increase would have a large JSD, but we might still want to call that transform "smooth" if those regions are spatially far from each other and the intermediate space varies smoothly from shrinking to stretching. On the other hand, it could be useful to describe transforms with less extreme values of the Jacobian determinant as "unsmooth" if changes from stretch to shrink occur over smaller spatial distances. In other words, it could be useful to describe *how quickly* (over space) transform changes occur, rather than describing *that* transform changes occur over all of space as JSD does. Next, we describe that the norm of the Hessian matrix could serve as a useful and complementary measure to the JSD.

**Table 4. Jacobian determinant standard deviation (JSD) and Hessian frobenius norm mean (HFM) where templates are sorted by descending JSD for a fixed algorithm (ANTs A).**

| Template | JSD | | | HFM | | |
|---|---|---|---|---|---|---|
| | ANTs A | CMTK A | Elastix A | ANTs A | CMTK A | Elastix A |
| JFRC2010 | 0.26 | 0.19 | 0.18 | 0.032 | 0.014 | 0.0029 |
| JFRC2013 | 0.23 | 0.26 | 0.23 | 0.039 | 0.017 | 0.013 |
| JRC2018F | 0.20 | 0.21 | 0.17 | 0.035 | 0.025 | 0.018 |
| FCWB | 0.47 | 0.32 | 0.23 | 0.051 | 0.038 | 0.0047 |
| Tefor | 0.26 | 0.23 | 0.14 | 0.040 | 0.019 | 0.0033 |

This algorithm has the capability of producing large deformations (i.e. it is not overly regularized), as by the large value of JSD for the FCWB template. The smallest average amount of deformation was obtained after registration to our template, JRC 2018F, followed closely by JFRC 2013.

The Hessian matrix is the matrix containing all partial second derivatives of a transformation, and describes how spatially quickly changes in the transformation occur. The Hessian of any linear transformation will be identically zero everywhere. Therefore the "size" of the matrix as measured by a matrix norm describes the extent to which a transformation is locally non-linear. The mean value of the Hessian matrix norm is an appropriate statistic for summarizing over space. We compare the Hessian matrix Frobenius norm mean (HFM) to the JSD in Section 3.2.3, and report both values in Table 4.

Table 4 shows the JSD and HFM for the five templates when using ANTs A as the registration algorithm (the best algorithm on average, measured by mean skeleton distance). This suggests that the JRC 2018F template may be "closer" on average to the shape of individual brains (in our testing cohort) in the sense that less deformation is required to transform those brains to the template. The conclusions for HFM are different, though. The JRC 2018 template is third smoothest on average according to HFM. This could be due to different tissue preparation methods used for evaluation brains and template building brains. The tissue preparation used for the JFRC 2010 and JFRC 2013 templates is more similar to the evaluation images than the preparation used for the images used to build our template.

To summarize, JSD and HFM give similar information regarding the spatial distortion of a transformation, though transformations with large JSD seem to yield qualitatively (visually) worse distortions than transformations with large HFM. We discuss this more in Supplementary Notes A.6 in S2 File. S4 Table in S2 File shows a complete listing of JSD and HFM statistics across templates and algorithms.

## 3.3 Computation time and supplementary results

Fig 6 plots mean skeleton distance against mean computation time (in CPU-hours). The choice of algorithm and parameters influences computation expense more than the choice of template. Our particular parameter choices for CMTK are generally faster than ANTs, and our parameter choices for elastix are about ten times faster still. We see that the best performing algorithm (ANTs A) is also the most computationally demanding. Finally, for the fastest algorithms (Elastix A and Elastix B) mean skeleton distance is smallest using the JRC 2018F template. JFRC 2013 is the next best template.

We also performed experiments varying the resolution of the two best templates JRC 2018F and Tefor when using the best-performing algorithm (ANTs A) (see Supplementary Notes A.4 in S2 File). The results show that a marked speed up in computation time can be achieved by registering downsampled images, for only a modest decrease in accuracy.

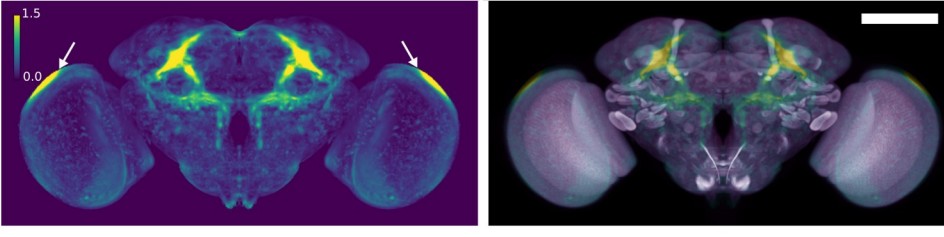

(A) Jacobian determinant map difference                    (B) Overlay

**Fig 7. A visualization of female vs male morphological differences.** (A) the maximum intensity projection (MIP) of the difference between male and female Jacobian determinant maps where values greater than 1 (yellow) indicate the male template is locally larger. (B) the MIP of the Jacobian determinant map overlayed with a MIP of the unisex template. Arrows indicate an artifact of this analysis in regions where the templates have no contrast, but the registration algorithm applies a strong deformation. Scale bar 100 μm.

## 3.4 Sexual dimorphism

Next, as a demonstration of efficacy, we briefly examine sexual dimorphism in the *Drosophila* brain, reproducing the results of Cachero et al. [60], using the brain templates we created. Consistent with those previous findings, we observe enlargement of anatomy associated with *fruitless* gene expression in the male JRC 2018 template relative to the female JRC 2018 template.

We compared the Jacobian determinant maps of two transformations: the transformation between the male and unisex template and that between the female and unisex template. These transformations should reflect the differences between the mean female and male shapes, respectively. Fig 7 shows the differences of the Jacobian determinant map for the transformations between the male-to-unisex and female-to-unisex templates. This qualitatively agrees with the result in Cachero et al. [60] that male enlarged regions correlate with *fruitless* gene expression.

## 4 Discussion

We created six *Drosophila* templates for female, male, and unisex central brains and ventral nerve cords. In Fig 1, we see that the unisex templates are generally more blurry than the single sex templates, likely because the unisex templates include inter-sex variation in addition to inter-individual variation. This is more pronounced for the VNC than for the central brain. Furthermore, male templates appear to be more blurry than the female templates both for the brain and VNC. This could be due to higher variability across male individuals, or poorer performance by registration algorithms for males, though we believe the former to be more likely.

It is difficult to determine the extent to which each aspect of template construction affects registration performance. Nevertheless, here we put our results in the context of other work attempting to estimate performance of various templates. In Fig 3, we see that the anatomical features in the *Drosophila* brain in our template are qualitatively more pronounced, having higher contrast than other existing templates. As a result, pixel-based similarity measures used by automatic registration algorithms may be more effective at optimization when the signal for the target image is less obscured by noise of different kinds (e.g., anatomical variability, imaging artifacts). This is potentially one of the reasons that the JRC 2018F template outperforms others.

### 4.1 Estimating registration quality

In this work we chose pairwise neuronal skeleton distance as the primary measure of registration performance. Most templates are not explicitly evaluated for registration accuracy at time

of publication [6, 7], though Arganda-Carreras et al. [8] use the overlap of anatomical labels to compare performance. Unlike reference image similarity, both skeleton distance and label overlap are relevant in the sense that these measures are not directly optimized for by the registration algorithm which can lead to meaningless results [56]. Both of these measures have advantages and drawbacks.

Label overlap is sensitive to errors in the spatial location of anatomical compartments that are (usually manually) labeled by human annotators. Therefore, it specifically focuses on regions that are of potential interest to researchers. However, human annotation includes arbitrary choices, and therefore this measure may miss errors in biologically relevant areas that annotators overlook. Furthermore, annotating many individual images can be costly and introduces the variability of manual labeling. On the other hand, having multiple individuals annotated is advantageous in that the anatomical labels themselves can also have uncertainty associated with them. Multiple labeled individuals enable multi-atlas segmentation [61], an extension of single-atlas segmentation, at the cost of increased computational cost. Another potential drawback of using label overlap is that only changes near the boundaries of labels effects the overlap measure, and so it cannot differentiate methods that perform differently at the interior of labels. Registration errors parallel to or along the boundary are also not distinguishable.

The skeleton distance measure we use in this work evaluates registration using automatically extracted anatomy *directly* rather than indirectly as labeled by human annotators and so avoids manual effort in obtaining a registration accuracy measure. It instead requires that an additional independent image channel be acquired and must deal with biological variability and algorithmic errors as confounding factors. Specifically, naïve skeletonization and pairwise matching rely on the assumption that the nearest points on two skeletons are in correspondence, and will result in under-estimates of distance. This also means that skeleton distance cannot differentiate errors parallel to the skeleton. Furthermore, any errors in the automatic segmentation of the skeletons will affect the distance computation. Biological variability could also be a significant limitation if this variability is much larger than the differences between templates/registration algorithms. In fact, we do observe a large variance in the distribution of skeleton distance for all templates and all algorithms. The errors we observe are consistent with estimates of registration accuracy / biological variability in previous studies (see Section 4.2). In this work, differences in the distributions of this measure between templates and algorithms were detectable, even in the presence of these sources of variability.

Currently, the JRC 2018 templates do not have anatomical labels of their own superimposed, except for those that are inherited implicitly through bridging transformations to other templates. As a result, the accuracy of these labels depends on the accuracy of bridging transformation. In future work, we plan to develop a set of anatomical labels in the space of the JRC 2018 templates.

## 4.2 Studies of registration accuracy and anatomical variability

In a previous study, Jefferis et al. [23] measured biological variability and registration accuracy by measuring the branch point of a projection neuron (PN) after entering the lateral horn, and estimated spatial variability of 2.64, 1.80, and 2.81 μm, for the three axes. This yields a mean euclidean distance of about 4.3 μm. Separately, they found that the mean axon positions of PNs within the inner antennocerebral tract (iACT) were 3.4 μm apart. In another study Peng et al. [14], developed a pointwise image registration method, "Brainaligner," and assessed its accuracy, as well as natural biological variability. They estimated the spatial variability of the axons in the iACT to 3.26 μm, quite similar to the 3.4 μm estimate in Jefferis et al. [23]. These

estimates, taken together are consistent with our estimate of a mean skeleton distance of about 4.0 μm for the best registration algorithm and the best template. This figure is slightly less than the 4.3 μm distance of the lateral horn PN branch point from and slightly greater than the 3.4 μm estimate of the iACT from Jefferis et al. [23]. The 3.26 μm estimate of Peng et al. [14] is lower still. We expect part of this difference has to do with the fact that their measure focuses on a single, perhaps easily localizable region, whereas we examine distances across the entire brain.

### 4.3 Deformation

As observed in Fig 5, one of the benefits of the JRC 2018 template is that an accurate registration can be obtained with less local deformation than for other brain templates. Note that we refer here to deformation introduced to the acquired image by registration. This is distinct from deformation to the anatomy caused by fixation, tissue preparation, and imaging. We discuss those effects briefly in Section 4.7. Less local deformation introduced by registration is beneficial because the morphology of anatomy is better preserved, and therefore the influence of the transformation is less likely to be a confounding factor in downstream analyses. For example, neuronal morphology is an important signal when searching across driver lines for a particular neuron or neurons of interest. This task benefits from normalizing spatial location (registration) and from comparing morphology. Therefore, minimizing deformation should improve neuron matching. For example, NBLAST [31] makes use of morphology of neurons in computing pairwise scores, and therefore any distortion in shape could prevent effective matching.

While the Jacobian- and Hessian-based measures of deformation are correlated, they do measure somewhat different properties of the transformation. The fact that some transformations with similar values of JSD can have markedly different values of HFM is evidence of this. Transformations produced by ANTs are less smooth than those produced by CMTK and elastix when measured by HFM.

The extent of deformation is just one of several factors to be considered when choosing a template and registration algorithm. A reasonable choice would be to accept larger deformations only if they produce more accurate registration results. A danger described in Rohlfing et al. [56] is that transformations with an unreasonably high degree of deformations can "trick" bad proxy measures of registration accuracy. Better measures of registration quality, such as the skeleton distance used here, can help to avoid this. For example, the ANTs C parameters were not regularized, and produced very unsmooth transforms, corroborated by high values of JSD in Fig 5. That algorithm also scored relatively poorly according to skeleton distance, which indicates that the measure is robust. This poor performance is visually apparent as well, as can be seen in the examples in Supplementary Notes C in S2 File.

In the following sections, we consider a few possible reasons for the improvements in performance we see when using JRC 2018 as a target for registration.

### 4.4 Influence of groupwise averaging

A potential concern in using an average of many individuals as a registration target is that some anatomical features are blurred or lost due to imperfect registration followed by averaging. In this case, it could be that the loss or blurring of these anatomical features removes a useful signal for the registration algorithm and could result in worse alignment. If this is true, then individual templates should perform better than average templates, since features are not blurred away. Our results suggest the opposite, that groupwise-average templates outperform

individual templates This agrees with other work on the human hippocampus from MRI [62], in ants [63], and *Drosophila* [8].

If an anatomical feature is blurred or lost in the process of template construction, then that feature must have been poorly aligned on average across the population, due perhaps to large anatomical variability. It could be that the presence of unreliable/ highly variable anatomy is useless or detrimental on average for registration across a large population. For example, highly variable anatomy could result in over-warping, especially when using algorithms with very flexible transformation models, since they can "force" improvements to the similarity measure even for incompatible anatomy, analogous to overfitting in machine learning. This is another possible reason that average templates outperform individual templates.

### 4.5 Symmetry

The central brain of *Drosophila* is largely left-right symmetric, but it does have a noteworthy asymmetry [64]. Any template created without enforcing symmetry would have small but widespread asymmetries caused by the particular specimens and images that contributed to the average, not *biologically meaningful* asymmetries. We therefore created a set of left-right symmetric templates. This has the disadvantage of removing *real* asymmetries from the template brain, but they can still be recovered by using appropriate analysis.

One approach is to compare image intensities in the left hemisphere to corresponding intensities in the right hemisphere using a "mirroring transformation." This technique requires an independent image channel, since registration will remove asymmetries from the image it uses. An advantage of symmetry is that it is straightforward to find the mirroring transformation mapping the left hemisphere to the right and vice versa. Manton et al. [30] and Schlegel et al. [32], for example, describe how such mirroring transforms are useful in performing neuronal comparisons across hemispheres, since many neurons in one hemisphere correspond to a partner in the other. Left-right morphological differences can be also recovered by analyzing the deformation fields for asymmetries, using deformation based morphometry [9]. Registration to a symmetric template will not affect analysis methods that probe for asymmetry using features unrelated to the intensity of the registration channel. For example, Linnweber et al. [65] count axons reconstructed from confocal images, and would not have been affected by registration to our symmetric template. As always, researchers must take care that their processing aligns with the goals of a study.

### 4.6 Influence of number of individuals to build atlas

Our templates were built using many more individuals than other average templates. The FCWB template averaged 26 individuals, and the Tefor brain averaged 10 individuals, where we average 36 female individuals, and 26 male individuals for the sex-specific but consider twice as many images by left-right flipping each individual. As a result, we are more confident that our templates are near the "mean shape" of the central brain for each sex. Arganda-Carreras et al. [8] explored how the number of individuals that comprised a template effects the registration quality to those templates. They found a plateau of performance for templates built using between seven and ten individuals. Yet, JRC 2018 generally outperforms Tefor. Next, we outline some of the potential reasons for this.

If the conclusion of Arganda-Carreras et al. [8] is true, then whatever performance improvement JRC 2018 achieves over the Tefor brain is not due to the number of individuals comprising the templates. We note first that the mean distance measures for JRC 2018 and Tefor are very similar when using the best performing but most costly registration algorithm (ANTs A). Using this algorithm, the average deformation energy (measured by Jacobian

determinant) is smaller for JRC 2018 (0.17) than for Tefor (0.23) (see Fig 5), suggesting that it is "closer" to the mean of our test subjects than the Tefor brain. This could also explain why JRC 2018 achieves better performance than Tefor when using registration methods that are more highly regularized (e.g. ANTs B and CMTK C). It could be the case that the template shape does not converge after only seven to ten individuals have been averaged, but that this shape change is not reflected in registration performance measures. It seems unlikely that image acquisition would affect the average deformation when registering to a template. Given that, our results support the conclusion that using more individuals for a template yields a more representative shape which reduces deformations for a given level of performance, or enables better performance for a smaller, fixed amount of deformation.

Finally, we note that one of the most widely used anatomical templates, the Allen Mouse Common Coordinate Framework, used many more individuals (1675) than we used in this work, and, as we did, also included left-right flips as well as the original images for a total of 3350 [3].

## 4.7 Influence of tissue staining, preparation, and genetics

Arganda-Carreras et al. [8] compare their Tefor template to the FCWB template [28] and show that Tefor has improved performance, a conclusion that our results corroborate. They suggest that the difference between tissue staining protocols could be a contributing factor in the observed performance improvement. Nc82 images were used both to build the Tefor brain and for testing in Arganda-Carreras et al. [8]. In our work, we also use nc82 images for testing, but instead use brp-SNAP tag for images that contributed to our template. Still, evidence suggests that nc82 images are more reliably registered to our template than to Tefor, despite the fact that images contributing to our template underwent different sample preparation and imaging, and that similar methodology (groupwise registration) was used to construct our template and Tefor. We would generally expect similar results when using either nc82 or SNAP tags, since their expressions are colocalized in the brain [47], though the SNAP labeling results in more even contrast throughout the brain. Our results suggest that a template built with images having differing but improved contrast is a better registration target than a template of the same modality but worse contrast.

The flies whose imagery was used to build the template had genotype brp-SNAP/+. We do not expect this choice of genetic background to pose a barrier to the usefulness of the template as a registration target in most cases, though mutants bred specifically for changes in brain morphology could be an exception. In this work, the flies used for evaluation were all from different genetic lines than those used in making the template. Janelia's Flylight project team has successfully registered over 234k brain samples from over 25k genetic lines, and over 101k VNC samples across over 24k genetic lines to JRC 2018.

Tissue preparation methods used will change the size and shape of the underlying anatomy as dehydration shrinks the samples [49]. It is difficult to assess changes more precisely, though often the goal of sample preparation is consistency across individuals. The availability of live imaging means it could soon be possible to generate a live whole-brain template that could be used to quantify these changes, but to date, no such study has been done to our knowledge. Qualitatively, we observe more spatial distortion for more lateral structures such as the optic lobes, consistent with Chiang et al. [22]. Similarly, the VNC undergoes more spatial change during preparation than the central brain due its longer and thinner shape.

## 4.8 Influence of template resolution

The resolutions of a brain template could affect the ability of registration algorithms to produce an accurate alignment. It could be that templates rendered at isotropic resolution could

outperform those with anisotropic voxels even when the moving image is anisotropic, as is the case with our testing data that were acquired with a confocal microscope. One of the aniso-tropic templates we evaluated here performs well in the best case (Tefor), while the other (FCWB) performs worse than isotropic templates generally.

We also explored how varying the resolution of the two best templates (JRC 2018 and Tefor) affects registration performance (see Supplementary Notes A.4 in S2 File). We conclude that it is possible for lower resolution templates to achieve similar performance to high resolu-tion templates with the benefit of computational savings. However, it is worth exploring different choices of algorithms in this case, since any given algorithm may not have uniform performance across different resolutions.

### 4.9 Accuracy/time tradeoff

The required level of accuracy of registration will vary from task to task. We recommend that researchers first experiment with the less computationally demanding algorithms/parameter sets, determine whether the results are adequate for their particular task, and if not, to try the potentially more accurate but more computationally demanding options. Specifically, among the parameter sets we tested, elastix is about ten times faster than the fastest parameter settings for CMTK and ANTs, and is therefore worth experimenting with first.

As a result, Fig 6 shows a cluster of templates with good performance that are fast to com-pute when using elastix. JRC 2018 is the best performing template in this cluster. Furthermore, the performance is also qualitatively much better using JRC 2018 than others when examining the registration results. Many examples of registration results are provided in Supplementary Notes C in S2 File.

## Supporting information

**S1 File.**
(PDF)

**S2 File.**
(TEX)

**S3 File.**
(PDF)

## Acknowledgments

The authors would like to thank Philipp Hanslovsky, Igor Pisarev, Alice Robie, Yoshi Aso, and Michael Reiser for helpful discussions, Zhihao Zheng for help with EM-LM registration analy-sis, Arnim Jenett and Ignacio Arganda-Carreras for the Tefor brain template, Greg Jefferis for incorporating our bridging transformations into the nat R library, and the anonymous reviewer for suggestions that improved and clarified the manuscript.

## Author Contributions

**Conceptualization:** John A. Bogovic, Stephan Saalfeld.

**Data curation:** John A. Bogovic, Hideo Otsuna, Masayoshi Ito, Geoffrey Meissner, Aljoscha Nern, Jennifer Colonell.

**Formal analysis:** John A. Bogovic.

**Funding acquisition:** Stephan Saalfeld.

**Methodology:** John A. Bogovic, Stephan Saalfeld.

**Resources:** Masayoshi Ito, Jennifer Jeter, Geoffrey Meissner, Aljoscha Nern, Jennifer Colonell, Oz Malkesman.

**Software:** John A. Bogovic, Hideo Otsuna, Larissa Heinrich, Stephan Saalfeld.

**Supervision:** Oz Malkesman, Kei Ito, Stephan Saalfeld.

**Visualization:** John A. Bogovic.

**Writing – original draft:** John A. Bogovic, Stephan Saalfeld.

**Writing – review & editing:** John A. Bogovic, Geoffrey Meissner, Stephan Saalfeld.

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
