## [Decision Letter · Decision Letter 0]

8 May 2020

PONE-D-19-36054

An unbiased template of the Drosophila brain and ventral nerve cord

PLOS ONE

Dear Dr. Bogovic,

Thank you for submitting your manuscript to PLOS ONE. After careful consideration, we feel that it has merit but does not fully meet PLOS ONE’s publication criteria as it currently stands. Therefore, we invite you to submit a revised version of the manuscript that addresses the points raised by reviewer #1. 

We would appreciate receiving your revised manuscript by Jun 22 2020 11:59PM. To enhance the reproducibility of your results, we recommend that if applicable you deposit your laboratory protocols in protocols.io, where a protocol can be assigned its own identifier (DOI) such that it can be cited independently in the future. For instructions see: http://journals.plos.org/plosone/s/submission-guidelines#loc-laboratory-protocols

We look forward to receiving your revised manuscript.

Kind regards,

Matthieu Louis

Academic Editor

PLOS ONE

Journal Requirements:

2. Thank you for including your funding statement; "This work was funded by HHMI."

Please expand the acronym “HHMI” (as indicated in your financial disclosure) so that it states the name of your funders in full.

3. Please ensure that you refer to Figure 13 in your text as, if accepted, production will need this reference to link the reader to the figure.

Reviewers' comments:

Reviewer's Responses to Questions

**Comments to the Author**

1. Is the manuscript technically sound, and do the data support the conclusions?

Reviewer #1: Yes

2. Has the statistical analysis been performed appropriately and rigorously? 

Reviewer #1: Yes

3. Have the authors made all data underlying the findings in their manuscript fully available?

Reviewer #1: Yes

4. Is the manuscript presented in an intelligible fashion and written in standard English?

Reviewer #1: Yes

5. Review Comments to the Author

Reviewer #1: This manuscript provides a state of the art standardized atlas of the Drosophila central nervous system (including the brain and the VNC) together with a detailed analysis of its quality and public open access availability in comparison to existing standardized brains. The analysis is of outstanding detail and quality, so that the standardized CNS will likely be a highly useful tool for the scientific community. The data are freely available at https://www.janelia.org/open-science/jrc-2018-brain-templates, and the format seems to be open source. The manuscript is well written and I have only minor suggestions for improvement.

1. It would be great to have a brief and simple description of how to use the atlas right up front in the manuscript. It is fantastic that the templates, tranformations, code, and descriptions are all publically available, but a brief paragraph within the manuscript that describes the use of these resources would likely attract many researcher to actually try it out.

2. It is understood that registration of genetically labeled neurons requires to have a standard of non-wildtype Drosophila. However, it might be useful to include a short paragraph of the origin of the respective genetic backgrounds. This would greatly help researchers, that are I not as experienced in Drosophila genetics as the authors are, to judge on potential differences /caveats when they will try to move forward and register data from their own work with different fly strains.

3. It seems a little harsh to state that the existing VNC atlas of Börner is not available at all. Although I have not made the effort to try to find it myself, I know people who have accessed it before. However, it is agreed that it is not as advanced as the one of this study, has not been analyzed nearly as deeply as the work presented, and most importantly, it is not useable with open access software which indeed significantly limits availability to the field. In addition to the high quality, being able to use open source code and software is a compelling advantage of the work presented.

4. A statement about the obvious tissues distortions, especially non-isometric shrinkage, as expected from the fixation, dehydration, and clearing protocols used for histology would be useful. Especially in the light of the expected increasing availability of live imaging data (without fixation etc.). What will users have to be aware of when trying to register functional data? (which would be great in the future) I am not asking for a rigorous analysis here, but a short paragraph provided by these true expert authors would be highly useful.

5. The term “irrelevant sources of variability” needs some additional explanation /justification, especially in the light of the recent study by Linneweber et al. (2020, Science. 367(6482):1112-1119.) which relates morphological variability to behavioral individuality.

6. I agree that the SD of the Jacobian determinant is a highly useful measure for non-isometric shrinkage, as explained nicely in section 4.2.2.. It seems that Hessian matrix norm also measures precisely this, and the reported values for JSD and HMN correlate nicely (table 8). Does this mean that both provide a similarly good means for judging on non-isometric tissue shrinkage? I understood the text like this, but admittedly had some difficulties understanding these sections. It would be nice to explain this somewhat clearer to the non-expert, and maybe also include some judgement on what values become nearly unacceptable for registration of future samples.

6. PLOS authors have the option to publish the peer review history of their article (what does this mean?). If published, this will include your full peer review and any attached files.

Reviewer #1: No

---

## [Author Response · Author response to Decision Letter 0]

22 Jun 2020

We thank the anonymous reviewer for their comments. We have addressed these in our revision and feel that the clarity and accessibility of the manuscript are improved as a result. Since our initial submission, we have made public a new transformation from our template to a newly released Drosophila electron microscopy dataset, called the "hemibrain". Our changes are highlighed in blue in the ``Revised Manuscript with Track Changes,'' with footnotes indicating the comment each change addresses. 

A summary of the changes are given below.

"It would be great to have a brief and simple description of how to use the atlas right up front in the manuscript. It is fantastic that the templates, transformations, code, and descriptions are all publicly available, but a brief paragraph within the manuscript that describes the use of these resources would likely attract many researcher to actually try it out."

We thank the reviewer for the suggestion and considering the accessibility of this work. We have added a brief new section at the end of the introduction (``Usage'') in which we describe the ways in which the resources we have created will be useful to various researchers in their own work.

"It is understood that registration of genetically labeled neurons requires to have a standard of non-wildtype Drosophila. However, it might be useful to include a short paragraph of the origin of the respective genetic backgrounds. This would greatly help researchers, that are not as experienced in Drosophila genetics as the authors are, to judge on potential differences /caveats when they will try to move forward and register data from their own work with different fly strains."

We have included a short discussion regarding the potential caveats when using this work with Drosophila of different genetic backgrounds, of which be believe there to be very few. As evidence for this, we point out our own internal success in registering many thousands of brains and VNCs from different genetic lines.

"It seems a little harsh to state that the existing VNC atlas of Borner is not available at all. Although I have not made the effort to try to find it myself, I know people who have accessed it before. However, it is agreed that it is not as advanced as the one of this study, has not been analyzed nearly as deeply as the work presented, and most importantly, it is not useable with open access software which indeed significantly limits availability to the field. In addition to the high quality, being able to use open source code and software is a compelling advantage of the work presented."

We echo the reviewer's appreciation of open data and software. We have softened our language slightly, but feel it is important to recognize this hurdle.

A statement about the obvious tissues distortions, especially non-isometric shrinkage, as expected from the fixation, dehydration, and clearing protocols used for histology would be useful. Especially in the light of the expected increasing availability of live imaging data (without fixation etc.). What will users have to be aware of when trying to register functional data? (which would be great in the future) I am not asking for a rigorous analysis here, but a short paragraph provided by these true expert authors would be highly useful.

We agree. Our revision includes a short discussion on this point, describing the expected kinds of tissue distortions due to fixation, dehydration, and clearing. We specifically point out the lack of ``ground-truth'' in this regard, and how novel technologies could help, as the reviewer points out.

"The term “irrelevant sources of variability” needs some additional explanation/justification, especially in the light of the recent study by Linneweber et al. (2020, Science. 367(6482):1112-1119.) which relates morphological variability to behavioral individuality."

The revised manuscript provides more details in the section 5.5 ("Symmetry") clarifying this point, including a reference to the recent related work pointed out by the reviewer. To summarize the additions, we more directly state how analysis can be done to recovery asymmetries after registration to our (symmetric) template, and what analysis to avoid when analyzing asymmetries.

"I agree that the SD of the Jacobian determinant is a highly useful measure for non-isometric shrinkage, as explained nicely in section 4.2.2.. It seems that Hessian matrix norm also measures precisely this, and the reported values for JSD and HMN correlate nicely (table 8). Does this mean that both provide a similarly good means for judging on non-isometric tissue shrinkage? I understood the text like this, but admittedly had some difficulties understanding these sections. It would be nice to explain this somewhat clearer to the non-expert, and maybe also include some judgement on what values become nearly unacceptable for registration of future samples."

We now more explicitly state our conclusions and recommendation in the main text in a single sentence. The supplement now includes some additional discussion and rationale regarding this topic.

---

## [Decision Letter · Decision Letter 1]

9 Jul 2020

An unbiased template of the *Drosophila* brain and ventral nerve cord

PONE-D-19-36054R1

Dear Dr. Saalfeld,

We’re pleased to inform you that your manuscript has been judged scientifically suitable for publication and will be formally accepted for publication once it meets all outstanding technical requirements.

Kind regards,

Matthieu Louis

Academic Editor

PLOS ONE

Additional Editor Comments (optional):

Reviewers' comments:

Reviewer's Responses to Questions

**Comments to the Author**

1. If the authors have adequately addressed your comments raised in a previous round of review and you feel that this manuscript is now acceptable for publication, you may indicate that here to bypass the “Comments to the Author” section, enter your conflict of interest statement in the “Confidential to Editor” section, and submit your "Accept" recommendation.

Reviewer #1: All comments have been addressed

2. Is the manuscript technically sound, and do the data support the conclusions?

Reviewer #1: Yes

3. Has the statistical analysis been performed appropriately and rigorously? 

Reviewer #1: Yes

4. Have the authors made all data underlying the findings in their manuscript fully available?

Reviewer #1: Yes

5. Is the manuscript presented in an intelligible fashion and written in standard English?

Reviewer #1: Yes

6. Review Comments to the Author

Reviewer #1: All of my comments have been fully addressed by the revisions made. The work will provide the scientific community with powerful and freely available tools for Drosophila brain atlas rmorphometry

7. PLOS authors have the option to publish the peer review history of their article (what does this mean?). If published, this will include your full peer review and any attached files.

Reviewer #1: No

---

## [Editor Report · Acceptance letter]

20 Jul 2020

PONE-D-19-36054R1 

An unbiased template of the *Drosophila* brain and ventral nerve cord 

Dear Dr. Saalfeld:

I'm pleased to inform you that your manuscript has been deemed suitable for publication in PLOS ONE. Congratulations! Your manuscript is now with our production department. 

Kind regards, 

on behalf of

Dr Matthieu Louis 

Academic Editor

PLOS ONE